

# Soil and stem xylem water isotope data from two pan-European sampling campaigns

Marco M. Lehmann[1,*], Josie Geris[2], Ilja van Meerveld[3], Daniele Penna[4], Youri Rothfuss[5], Matteo Verdone[4], Pertti Ala-Aho[6], Matyas Arvai[7], Alise Babre[8], Philippe Balandier[9], Fabian Bernhard[1], Lukrecija Butorac[10], Simon D. Carrière[11], Natalie C. Ceperley[12], Zuosinan Chen[6], Alicia Correa[13], Haoyu Diao[14], David Dubbert[15], Maren Dubbert[15], Fabio Ercoli[16], Marius G. Floriancic[17], Teresa E. Gimeno[18], Damien Gounelle[19], Frank Hagedorn[1], Christophe Hissler[20], Frédéric Huneau[21], Alberto Iraheta[22], Tamara Jakovljević[23], Nerantzis Kazakis[24], Zoltan Kern[25], Karl Knaebel[26], Johannes Kobler[27], Jiri Kocum[28], Charlotte Koeber[15], Gerbrand Koren[29], Angelika Kübert[30], Dawid Kupka[31], Samuel le Gall[5], Aleksi Lehtonen[32], Thomas Leydier[21], Philippe Malagoli[9], Francesca Sofia Manca di Villahermosa[4], Chiara Marchina[33], Núria Martínez-Carreras[20], Nicolas Martin-StPaul[19], Hannu Marttila[6], Aline Meyer Oliveira[3], Gael Monvoisin[34], Natalie Orlowski[35], Kadi Palmik-Das[16], Aurel Persoiu[36], Andrei Popa[37], Egor Prikaziuk[38], Cécile Quantin[34], Katja T. Rinne-Garmston[39], Clara Rohde[15], Martin Sanda[40], Matthias Saurer[14], Daniel Schulz[5], Michael P. Stockinger[26], Christine Stumpp[26], Jean-Stéphane Vénisse[9], Lukas Vlcek[28], Stylianos Voudouris[41], Björn Weeser[13], Mark Wilkinson[42], Giulia Zuecco[33], Katrin Meusburger[1]

*Correspondence to*: Marco M. Lehmann (marco.lehmann@wsl.ch)

[1]Forest Soils and Biogeochemistry, Swiss Federal Institute for Forest, Snow and Landscape Research WSL, Birmensdorf, Switzerland

[2]School of Geosciences, University of Aberdeen, Aberdeen, United Kingdom

[3]Department of Geography, University of Zurich, Zurich, Switzerland

[4]Department of Agriculture, Food, Environment and Forestry (DAGRI), University of Florence, Florence/Firenze, Italy

[5]Institute of Biogeosciences, Agrosphere (IBG-3), Forschungszentrum Jülich GmbH, Jülich, Germany

[6]Water, Energy and Environmental Engineering Research Unit, University of Oulu, Oulu, Finland

[7]Institute for Soil Sciences, HUN-REN Centre for Agricultural Research, Budapest, Hungary

[8]Faculty of Science and Technology, University of Latvia, Riga, Latvia

[9]Université Clermont Auvergne, INRAE, UMR PIAF, Clermont-Ferrand, France

[10]Department of Forestry, Institute for Adriatic Crops and Karst Reclamation, Split, Croatia

[11]UMR METIS, Sorbonne Université, UPMC, CNRS, EPHE, Paris, France

[12]Hydrology Group, Institute of Geography & Oeschger Centre for Climate Change Research, University of Bern, Bern, Switzerland

[13]Centre for International Development and Environmental Research (ZEU), Justus Liebig University Giessen, Germany

[14]Forest Dynamics, Swiss Federal Institute for Forest, Snow and Landscape Research WSL, Birmensdorf, Switzerland





[15]Isotope Biogeochemistry and Gas Fluxes, Leibniz Centre for Agricultural Landscape Research (ZALF), Müncheberg, Germany

[16]Chair of Hydrobiology and Fisheries, Institute of Agricultural and Environmental Sciences, Estonian University of Life Sciences, Tartu, Estonia

[17]Department of Civil, Environmental and Geomatic Engineering, ETH Zürich, Zürich, Switzerland

[18]CREAF, Bellaterra, Spain

[19]URFM, INRAE, Domaine Saint Paul, Site Agroparc, Avignon, France

[20]Catchment and Ecohydrology group, Environmental Sensing and Modelling unit, Luxembourg Institute of Science and Technology, Belvaux, Luxembourg

[21]CNRS UMR 6134 SPE, Université de Corse, Corte, France

[22]Institute for Geoecology, TU Braunschweig, Braunschweig, Germany

[23]Division for Forest Ecology, Croatian Forest Research Institute, Jastrebarsko, Croatia

[24]Laboratory of Hydrogeology, Department of Geology, University of Patras, Faculty of Natural Sciences, Rion, Patras, Greece

[25]Institute for Geological and Geochemical Research, HUN-REN Research Centre for Astronomy and Earth Sciences, Budapest, Hungary

[26]Department of Water, Atmosphere and Environment, Institute of Soil Physics and Rural Water Management, University of Natural Resources and Life Sciences, Vienna, Austria

[27]Ecosystem Research & Environmental Information Management, Environment Agency Austria, Vienna, Austria

[28]Institute of Hydrodynamics, Czech Academy of Sciences, Prague, Czech Republic

[29]Copernicus Institute of Sustainable Development, Utrecht University, Utrecht, Netherlands

[30]Institute for Atmospheric and Earth System Research / Physics, University of Helsinki, Helsinki, Finland

[31]Department of Forest Ecology and Silviculture, Faculty of Forestry, University of Agriculture in Kraków, Poland

[32]Natural Resources Institute Finland (Luke), Helsinki, Finland

[33]Department of Land, Environment, Agriculture and Forestry, University of Padova, Legnaro, Italy

[34]Université Paris-Saclay, UMR8148 GEOPS, Orsay, France

[35]Chair of Forest Sites and Hydrology, Institute of Soil Science and Site Ecology, TU Dresden, Tharandt, Germany

[36]Emil Racovita Institute of Speleology, Romanian Academy, Cluj-Napoca, Romania and Stable Isotope Laboratory, Stefan cel Mare University, Suceava, Romania

[37]National Institute for Research and Development in Forestry "Marin Dracea", Bucharest, Romania

[38]Faculty of Geo-Information Science and Earth Observation (ITC), University of Twente, Enschede, Netherlands

[39]Stable Isotope Laboratory of Luke (SILL), Natural Resources Institute Finland (Luke), Finland

[40]Department of Landscape Water Conservation, Faculty of Civil Engineering, Czech Technical University, Prague, Czech Republic

[41]Earth Sciences and Environmental Technologies Division, IFP Energies Nouvelles, Rueil-Malmaison, France

[42]Environmental and Biochemical Sciences, James Hutton Institute, Aberdeen, United Kingdom



**Abstract.** Stable isotope ratios of hydrogen ($\delta 2H$) and oxygen ($\delta 18O$) are crucial for studying ecohydrological dynamics in forests. However, most studies are confined to single sites, resulting in a lack of large-scale isotope data for understanding tree water uptake. Here, we provide a first systematic isotope dataset of soil and stem xylem water collected during two pan-European sampling campaigns at 40 beech (Fagus sylvatica), spruce (Picea abies), or mixed beech-spruce forest sites in spring and summer 2023 (Lehmann et al., 2024). The dataset is complemented by additional site-, soil-, and tree-specific metadata. The samples and metadata were collected by different researchers across Europe following a standardized protocol. Soil samples were taken at up to 5 depths (ranging from 0 to 90 cm) and stem xylem samples from three beech and/or spruce trees per site. All samples were sent to a single laboratory, where all analytical work was conducted. Water was extracted using cryogenic vacuum distillation and analyzed with an isotope laser spectrometer. Additionally, a subset of the samples was analyzed with an isotope ratio mass spectrometer. Data quality checks revealed a high mean total extraction efficiency, mean absolute water amount (> 1 mL), as well as high analytical accuracy and precision. The water isotopic signature of soil and stem xylem water varied as a function of the geographic origin and changed from spring to summer across all sites. While $\delta^2H$ and $\delta^{18}O$ values were strongly correlated, the soil water data plotted closer to the Global Meteoric Water Line (GMWL) than the stem xylem water. Specifically, the $\delta^2H$ values of the stem xylem were more enriched than those of the soil water, leading to a systematic deviation from the GMWL. Isotopic enrichment of the stem xylem water was larger for spruce than for beech trees at mixed forest sites. This dataset is particularly useful for large-scale studies on plant water use, ecohydrological model testing, and isotope mapping across Europe.

**Keywords:** Critical Zone Science, Europe, Forest, Hydrology, Hydrogen Isotopes, Oxygen Isotopes, Root Water Uptake, Soil Water Recharge, Water Stable Isotopes, Water Sources.

## 1    Introduction

Understanding how tree water uptake from soils varies with species, site characteristics, time, and across climate zones is essential to assess forest resilience to climate change; particularly the response of forests to the increasing frequency and intensity of droughts (Lindner et al., 2010; Spinoni et al., 2014; Büntgen et al., 2021). Despite some uncertainties, the stable isotope ratios of hydrogen ($\delta^2H$) and oxygen ($\delta^{18}O$) in water extracted from soil and plants allow for the determination of the sources of water that are used by plants and to quantify the relative contribution of different water sources to plant water use (Rothfuss and Javaux, 2017; Beyer and Penna, 2021). Determination of water uptake patterns based on isotope data assumes that roots do not discriminate against the heavier hydrogen and oxygen stable isotopes during water uptake (Poca et al., 2019). Additionally, it is assumed that: (i)  the sampling design captures the spatiotemporal variability of the isotopic composition of soil water sources, (ii) the water extracted from the plant xylem is a mixture of the different water sources taken up from the soil profile without isotopic alteration (e.g., due to stem evaporation or leaf transpiration, see Ellsworth and Sternberg (2015), and (iii) soil and xylem samples are collected, transported, stored, and extracted in a manner that avoids isotope fractionation (Ceperley et al., 2024). Although these assumptions are not always met, the method described here—





whether used independently or in combination with others—can effectively test our understanding of the
mechanisms driving plant responses to both short- and long-term droughts. It is also now affordable enough for
practical applications beyond the field of isotope ecohydrology (Penna et al., 2018). Isotope-based analyses in
forest ecosystems have, for example, been used to determine the changes in root water uptake depths of trees in
response to drought (Brinkmann et al., 2018; Gessler et al., 2022), whether trees use summer or winter precipitation
(Allen et al., 2019; Floriancic et al., 2024a), soil water, groundwater, or streamwater (Bowling et al., 2017; Engel
et al., 2022), or to assess competitive or complementary water use strategies (Penna et al., 2020; Kinzinger et al.,
2024). However, systematic datasets at large scales, i.e., spanning continents or multiple countries, are lacking.
This hampers our understanding of how water uptake strategies for the same tree species vary across space and
time (Beyer and Penna, 2021; Orlowski et al., 2023; Dubbert and Werner, 2019; Bachofen et al., 2024).
There are established networks for the observation of isotopes in freshwater systems, such as precipitation by the
International Atomic Energy Agency (IAEA) Global Network of Isotopes in Precipitation (GNIP), which currently
contains data for 300 active sites in 93 countries (Terzer-Wassmuth et al., 2023). The Global Network of Isotopes
in Rivers (GNIR) contains data from 750 sites in 35 countries (Halder et al., 2015). Both networks have proven to
provide valuable  input data for modeling of the local to regional climate or surface-atmosphere water interactions
with process-based (e.g., CLM, Wong et al. (2017), ISOLSM Cai et al. (2015), ECHAM5-JSBACH Haese et al.
(2013)) or statistical models (e.g., Isoscapes (Bowen, 2010; Terzer et al., 2013; Allen et al., 2018; Koeniger et al.,
2022), and time series analyses (Nelson et al., 2021; Erdélyi et al., 2023; Reckerth et al., 2017). They have
furthermore helped to assess water flow pathways and the fraction of young water in streamflow (Von Freyberg
et al., 2018; Floriancic et al., 2024b). The Moisture Isotopes in Biosphere and Atmosphere (MIBA) network,
initiated by the IAEA in 2003-2004, is, to our knowledge, the only international network to survey the isotopic
composition of water across different ecosystem compartments (i.e., soil, plant stems and leaves, soil, and
atmospheric vapor). However, despite the global distribution of sites at the time of the establishment and a local
application in Australia (Twining et al., 2006), the network is currently inactive.
Building on the idea of the MIBA and the proven usefulness of national large-scale sampling campaigns to
determine regional differences in tree water uptake (Allen et al., 2019), the COST Action "WATer isotopeS in the
critical zONe: from groundwater recharge to plant transpiration WATSON" (CA19120) organized two sampling
campaigns across Europe in 2023. The effort took advantage of the European network of researchers to establish
a unique systematic water isotope dataset and corresponding metadata. More specifically, the goal of the sampling
campaigns was to obtain soil and stem xylem water isotope data of two tree species, namely beech (*Fagus sylvatica*
L.) and spruce (*Picea abies* (L.) H. Karst) across a large climate gradient for the spring (25[th] May to 16[th] June) and
summer (17[th] August to 18[th] September) of 2023. The two time points were selected to compare tree water uptake
patterns under different soil moisture conditions (e.g., lower soil moisture in summer). The two species were
selected because of their wide geographical distribution across Europe (Figure 1) and their important ecological
and economical relevance, as well the expected differences in water uptake depth (Allen et al. 2019; Brinkmann
et al. 2018; Goldsmith et al. 2019) with beech having a deeper rooting system than spruce.
During the European sampling campaigns, a total of 381 soil and 311 stem xylem samples were taken from 40
sites across 18 countries, following a standardized protocol. The water of these samples was cryogenically
extracted and isotopically analyzed in a single laboratory. The simultaneous collection of soil and stem xylem



samples across all European sites, combined with a centralized processing of the samples, ensures the uniqueness of this dataset. Using one laboratory prevents inconsistencies that might arise from varying sample handling and analysis methods, which can influence isotopic offsets (Orlowski et al., 2016; Orlowski et al., 2018). The isotope dataset is accompanied by site-, soil-, and tree-specific metadata at each location. Together, the metadata and isotope data provide a strong foundation for research on tree water use, model testing, and isotope mapping. This manuscript outlines the sample collection process, cryogenic water extraction, and isotope analysis, and details the dataset organization and metadata. Finally, we give an overview of the data and discuss potential applications. The full dataset is freely available (Lehmann et al., 2024).

## 2 Material and Methods

### 2.1 Organization of the WATSON pan-European sampling campaigns

During the initial phase (spring 2023), the members of the WATSON community (~200 members at that time) were contacted to assess their interest in participating in a coordinated sampling campaign. Based on the large interest, a core team was formed. The core team asked researchers from a similar region to form one team to keep the laboratory and analytical work manageable, while still obtaining samples from a broad geographic region. The core team wrote detailed instructions to ensure systematic sampling. The instructions provided detailed standardized protocols for collecting soil and stem xylem samples, including specifications for sampling depths, core dimensions and numbers, and the maximum number of samples. The protocols also covered short-term sample storage and shipment to the Swiss Federal Institute for Forest, Snow, and Landscape Research in Birmensdorf, Switzerland (WSL Birmensdorf), where all cryogenic water extractions and isotopic analyses were performed. In addition, participants were given instructions on how to take pictures for canopy cover analysis and the list of required metadata (e.g., geographical parameters, soil properties, tree diameter and height). The instructions were emailed to all interested contributors prior to the first sampling campaign in spring 2023 (Section S1). For the second campaign in summer 2023, the sampling protocol was slightly updated for clarity (i.e., weather conditions at sampling day, bark removal during stem xylem sampling, labelling of exetainers, taking photos) and emailed to all interested contributors again (Section S2). In addition, we held an online meeting between the two sampling campaigns to provide feedback to the participants, clarify any field issues, and answer questions.

### 2.2 Description of the sampling sites

Samples were taken from 40 different mono-specific and mixed forest sites with beech trees (*Fagus sylvatica*; 14 sites), spruce trees (*Picea abies;* 13 sites), or both tree species (13 sites) in 18 European countries (Figure 1; Table 1): 36 sites were sampled in the spring and 39 sites in the summer. For 35 of the 40 sites, samples were collected during both campaigns. In three of the sampling sites, separate beech (LIZ1, GLS1, WEI1) and spruce (LIZ2, GLS2, WEI2) stands were found close to each other (i.e. the sampling sites share the same geographic coordinates). Although there was a good cover of sites across central Europe for both species, most north-eastern sites were sampled for spruce only, while the spread of sampled beech trees extended more to south-western Europe. The sampling sites correspond to the natural and naturalised ranges of the tree species across Europe (Figure 1) and cover a range of temperate (Köppen-Geiger Cfa, Cfb, Csb) and cold (Köppen-Geiger Dfb, Dfc) climates. The

sampling sites also differed in elevation (14 to 1870 m a.s.l.; Table 1). The sampling sites were evenly distributed
across different slopes (i.e., flat, gentle, and steep). Most sites were located on Cambisols or Leptosols; with just
one Histosol (i.e., peat at the site ROT in Finland). The maximum existing soil depth varied between 0.3 m and >
1 m and for half of the sites, the maximum soil depth was > 0.6 m. Canopy cover was determined for 30 of the 40
sampling sites from non-hemispherical photographs taken with a phone camera, as described in Section S3. Most
of the pictures were taken during the spring campaign, however, for some sites, pictures were taken during the
summer campaign or both campaigns. For the sites for which canopy cover could be determined, it was generally
higher for the beech trees than the spruce trees (Table 1).
**Table 1:** Summary statistics for sampling campaigns across 40 European beech and spruce study sites, including
13 sites with both species. *Köppen-Geiger classification based on Beck et al. (2023).

| | | Beech | Spruce |
|---|---|---|---|
| Number of sites | | 27 | 26 |
| Number of sites sampled during both campaigns | | 24 | 23 |
| Elevation [m a.s.l.] | Min | 63 | 14 |
| | Mean | 756 | 648 |
| | Max | 1541 | 1870 |
| Climate* (Köppen-Geiger classification) [number of sites] | Cfa | 1 | 0 |
| | Cfb | 10 | 6 |
| | Csb | 1 | 0 |
| | Dfb | 14 | 14 |
| | Dfc | 1 | 6 |
| Tree height [m] | Min | 7 | 4 |
| | Mean | 22 | 23 |
| | Max | 44 | 39 |
| Diameter at breast height [cm] | Min | 11 | 8 |
| | Mean | 39 | 36 |
| | Max | 87 | 65 |
| Canopy cover (%) | Min | 58 | 54 |
| | Mean | 88 | 80 |
| | Max | 100 | 94 |

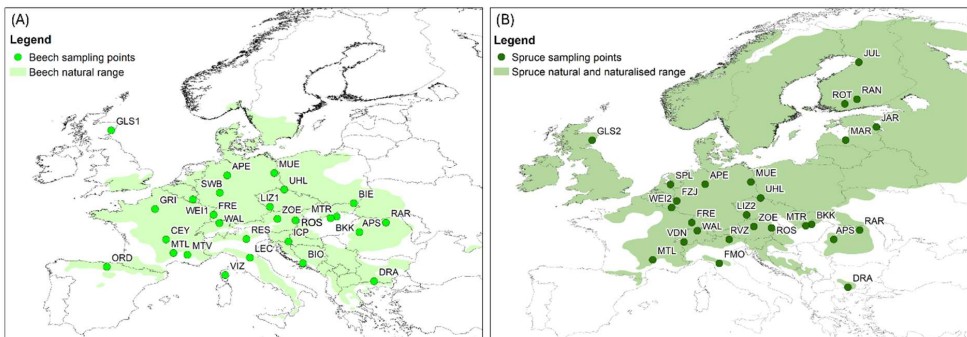

**Figure 1.** Maps showing the sampling sites (circles) for beech (A) and spruce (B) trees and their natural and naturalised ranges across Europe (shaded areas; data from Caudullo et al. (2017)).

### *2.3    Sampling, transport, and storage of stem xylem and soil samples*

At each sampling site, three beech (*Fagus sylvatica*) and/or three spruce (*Picea abies*) trees were selected based on their representativeness for the stand. The selected spruce and beech trees ranged in size but were similar in mean height (22-23 m) and diameter at breast height (36-39 cm, Table 1). Stem xylem samples were taken from each selected tree at breast height using a 0.5 cm increment borer. Each sample (one per selected tree) consisted of two to three ~5 cm long stem sapwood samples. Most samples consisted of fully intact wood cores; but 9.8% of all stem xylem samples were non-intact stem xylem samples. The outer and inner bark of the wood cores were removed from the cores, yet, bark residue was observed in 40% of all stem xylem samples after cryogenic water extraction. The same three trees were sampled during both campaigns at each site, except at the beech site GRI, where different trees were sampled in spring and summer, and at the beech site MTV, where six samples were taken. This resulted in a total of 311 stem xylem samples.

In addition to the stem xylem samples, soil samples were taken at each site for each sampling campaign with a manual soil auger. The samples were typically taken from one soil core at three to five depths spanning 10 cm intervals (0-10, 10-20, 20-30, 50-60, and 80-90 cm below the surface), but occasionally also for other depths. The number of soil samples and the depth of the deepest soil sample depended on the maximum existing soil depth at the sampling site. The soil samples were taken from a location close to the selected trees. The litter was removed before taking the 0-10 cm soil sample. For some sites and sampling campaigns, soil samples from an additional two to four soil cores were taken. For a few sites with both species (i.e., DRA, FRE, UHL, ZOE), soil cores were separately taken for beech, spruce, and both species. This resulted in a total of 381 soil samples.

Stem xylem and soil samples were transferred into 12 mL gas-tight glass vials ("Exetainers", Labco, Lampeter, UK). For the soil samples, exetainers were filled with 50-80% of their volume with soil. Some soil and stem xylem samples (13% of all 692 samples) were stored in other types of gas-tight plastic or glass vials. Most samples were taken midday on dry and sunny days. Samples were handled as fast as possible to avoid evaporative fractionation. Back in the laboratory, all samples were stored in a refrigerator to avoid moisture loss to evaporation and subsequent isotope fractionation until transportation. All samples were then shipped without cooling and arrived



within four weeks of the final day of each sampling campaign at the laboratory at WSL Birmensdorf in
Switzerland, where they were kept at -20°C until cryogenic water extraction.
**2.4    Cryogenic vacuum water extraction**
Water was extracted from all 692 samples at WSL Birmensdorf using a cryogenic vacuum distillation method as
described in Diao et al. (2022). In brief, the exetainers with the samples were taken from the freezer and fitted with
polypropylene fiber filters (Nozzle protection filter, Socorex Isba SA, Ecublens, Switzerland) to prevent particles
from being drawn into the extraction line. Samples originally stored in other types of vials were transferred to
exetainers that fit the cryogenic vacuum distillation system. Samples were then heated to 80°C in a water bath,
while the extraction line was kept under a vacuum of < 5 Pa (BS2212, Brook Crompton Ltd, Doncaster, UK). The
extracted water was trapped in U-shaped glass tubes, constantly kept in liquid nitrogen. After a minimum of 2
hours, the water extraction was stopped and atmospheric pressure was established in the extraction line by passing
dry nitrogen gas through it. Then, the U-tubes were removed, the ends of the tubes were closed with rubber plugs
and the water samples were thawed at room temperature. Depending on the extracted water amount, the water was
pipetted to 350 μL or 2 mL glass vials (Infochroma AG, Goldau, Switzerland) and kept frozen at -20°C until
isotope analysis. A few samples that appeared turbid after extraction were filtered with 0.45 μm nylon syringe
filters (Infochroma AG).
We determined the sample weight before water extraction ("fw"), after water extraction ("dw1"), and after drying
at 105°C for 24 hours (dw2) to estimate the absolute water amount ("awa"), the total extraction efficiency ("tef"),
and the gravimetric water content (gwc) for each sample (for equations, see Table 3). The sample weights (i.e.,
"fw", "dw1", "dw2") were corrected for the weight of the exetainer ("exe_weight", Table 3). The latter was based
on the mean weight of approximately thirty exetainers for 10 different types ("exe_type", Table 3; i.e., different
combinations of glass vials, cap with a rubber seal, and label), which averaged around 13.0 g and varied by a
maximum of 0.3 g. Across all soil and stem xylem samples (Figure 2A), "awa" averaged around 1.4 mL, and was
well above the critical thresholds for extracted water volume in the vast majority of samples (Diao et al., 2022).
The average value for "tef" was 100.6%, and was for most samples (N = 543) within the optimal range of 98-
102% (Ceperley et al., 2024). The "gwc" varied between soil samples and stem xylem samples of beech and spruce,
averaging around 40.9%, 61.3%, and 83.9%, respectively (Figure 2C). Note that variations in "awa", "tef", and
"gwc", and "tef" values > 100%, may partly be due to uncertainties arising from the estimation of the exetainer
weight ("exe_weight"; Table 3), reflecting an average value rather than the actual weight of each exetainer.

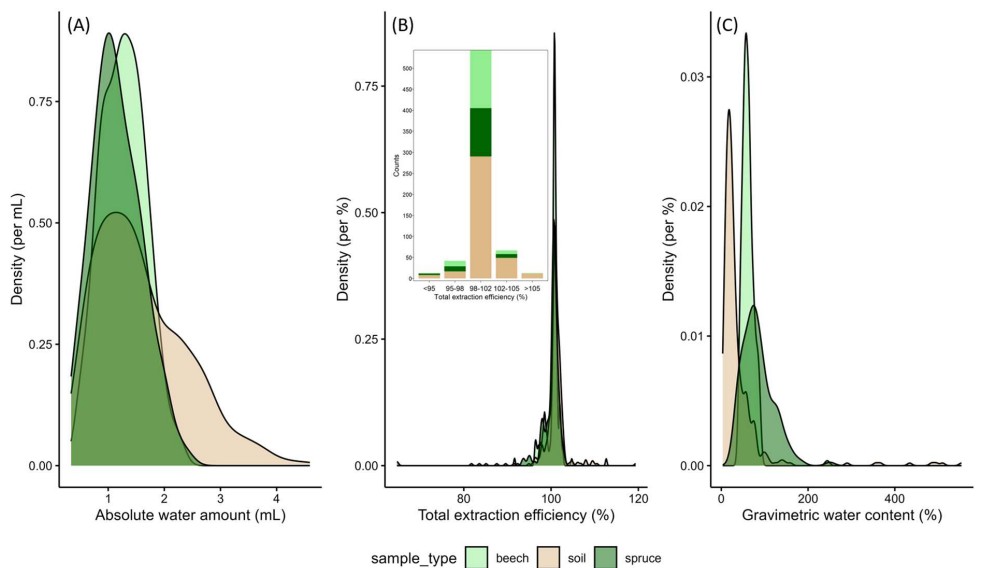

**Figure 2:** Density plots for (A) the extracted absolute water amounts, (B) the total extraction efficiency (tef), and
(C) the gravimetric water content (gwc) for stem xylem (beech and spruce) and soil samples for all samples
analysed (i.e., all sites and sampling campaigns). The insert in figure (C) shows the sample count for different
types of samples across five different tef classifications.

### 2.5 Isotope analysis with laser spectrometer and IRMS

The stable isotope ratios of hydrogen ($\delta^2H$) and oxygen ($\delta^{18}O$) of the cryogenically extracted water were measured
at WSL Birmensdorf using a cavity ring-down spectrometer (L2140i, Picarro Inc., Santa Clara, USA) connected
to a micro-combustion module (MCM) to eliminate sample artefacts caused by co-extracted organic compounds
(Martín-Gómez et al., 2015). Each sample was injected eight times and the average of the final five injections was
taken to minimize memory effects (Penna et al., 2012). Samples were calibrated with four reference isotope
standards spanning from -10.5‰ to -120.2‰ for $\delta^2H$ and from -3.0‰ to -16.1‰ for $\delta^{18}O$ (LGR; Envitec NV,
Lessines, Belgium) and normalized to the international Vienna Standard Mean Ocean Water (VSMOW-2) scale.
The maximum deviation (i.e., accuracy) of an interspersed in-house laboratory standard (analysed every ~25
samples, $\delta^{18}O$: -9.6‰, $\delta^2H$: -84.9‰) from the expected value was ≤ 0.2‰ for $\delta^{18}O$ and ≤ 0.5‰ for $\delta^2H$. The
standard deviation (SD) of the repeated measurements of the laboratory standards (i.e., precision) was ≤ 0.1‰ for
$\delta^{18}O$ and ≤ 0.6‰ for $\delta^2H$.

To check for spectral interferences with plant-produced volatile organic compounds during the isotope analysis
with laser spectrometer, a subset of 83 samples were also analyzed using a thermal combustion/elemental analyzer
(TC/EA) coupled to a DeltaPlus XP isotope ratio mass spectrometer (IRMS, Finnigan MAT, Bremen, Germany),
with a typical precision of 1.0‰ for $\delta^2H$ and 0.2‰ for $\delta^{18}O$. This subset was representative for both sampling
campaigns, sample types (stem xylem vs. soil), tree species, geographic locations, and range of isotopic values.
The IRMS data were highly correlated with the data of the laser spectrometer (Figures 3A, 3B). Most of the data



were within the range of ± 1 SD and showed a positive offset for both elements (Figure 3C). The $\delta^2$H and $\delta^{18}$O

offset between the two types of analysis had mean values around 0.7‰ and 0.3‰ across all samples (Figure 3C),

respectively. These mean offsets represent the average of the differences between the two methods, accounting for

both positive and negative values. The SD of these offsets were 1.4‰ for $\delta^2$H and 0.5‰ for $\delta^{18}$O, indicating the

variability around the mean offsets, not zero. Paired t-tests across the samples of the subset show that the $\delta^2$H and

$\delta^{18}$O differences between the two analytical methods were significantly ($P < 0.05$) larger for spruce (mean = 0.7‰

and 1.1‰) than for beech (mean = 0.4‰ and 0.7‰) and soils of all depth (only significant for $\delta^2$H; mean = 0.6‰).

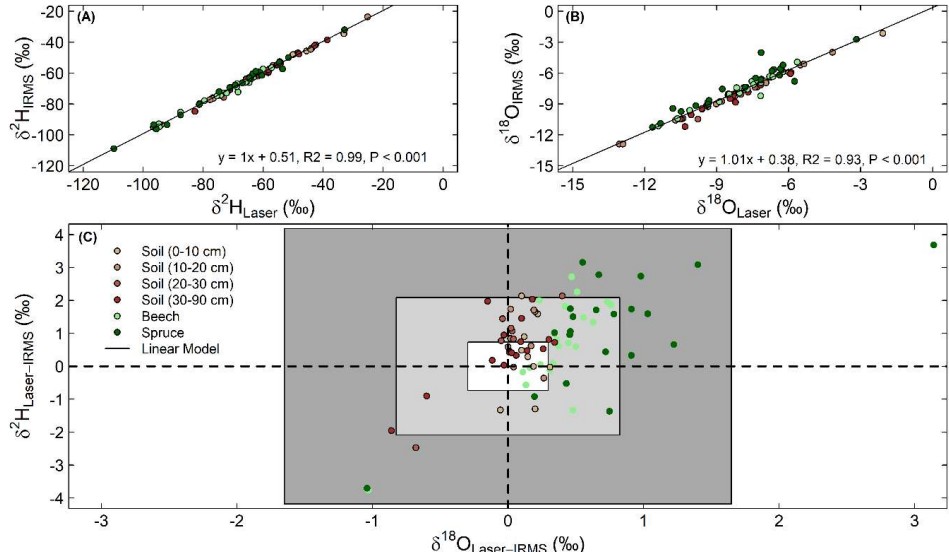

**Figure 3:** Linear relationships between hydrogen (A; $\delta^2$H) and oxygen (B; $\delta^{18}$O) isotopic composition for the water
samples analyzed using a laser spectrometer (Laser) and an isotope ratio mass spectrometer (IRMS). Panel (C)
displays a biplot of the differences in the $\delta^{18}$O and $\delta^2$H values for the two instruments. The small white box in the
middle of C represents the mean isotopic difference, while the light grey and dark grey boxes denote ± one and
two standard deviations for the isotopic difference, respectively.

## 2.6 Description of the dataset

The dataset consists of three comma-separated files and one zip file with photos of the canopy at the sampling
sites. The first datafile ("WATSON_Metadata.csv") contains all the metadata about the sampling sites including
site-, soil- and tree-specific information (Table 2), the second file ("WATSON_Isotopedata.csv") contains the
information about sample weights, cryogenic water extraction and the actual hydrogen and oxygen isotope data
(Table 3), and the third file ("WATSON_Canopydata.csv") contains the information on the canopy cover (Table
4). The photos on which the canopy cover data are based are stored in the "WATSON_Canopy_Pictures.zip" file.
All files can be linked by the "site_id", which is a three-letter identifier of the sampling sites.

**Table 2:** Description of the columns in the "WATSON_Metadata.csv" file containing all the meta-information
about the sampling sites [and units].

| Column name | Description |
| --- | --- |
| site_id | A three-letter identifier of the sampling site. Note that for the three sites (LIZ, GLS, WEI), an additional number was added indicating the species: "1" refers to beech and "2" to spruce. |
| site_name | Full site and country name |
| country_id | A two-letter country code, as defined in ISO 3166-1 |
| latitude | Latitude in decimal degree rounded to three decimals, WGS84 coordinate system |
| longitude | Longitude in decimal degree rounded to three decimals, WGS84 coordinate system |
| elevation | Elevation of the sample site [m above sea level] |
| slope_type | Descriptor of the slope: "flat", "gentle" or "steep" |
| spruce_site | Descriptor highlighting whether spruce trees were sampled at the site ("yes") or not ("no"). |
| beech_site | Descriptor highlighting whether beech trees were sampled at the site ("yes") or not ("no"). |
| stand_type | Descriptor highlighting whether the stand is a mixed species stand ("mixed") or a monoculture stand ("mono"). Note that "mixed" refers to stands with various species, not limited only spruce and beech. |
| understory | Descriptor highlighting the presence of understory vegetation ("yes") or not ("no"). |
| soil_type | Soil type according to the FAO classification |
| soil_texture | Soil texture based on either measurement of the sand, silt and clay content or hand tests in the field (see Section S1, S2). |
| soil_depth_max | Maximum soil depth [m], for soils deeper than 1 m, > 1 is used. |
| sampling_doy_spring | Day of the year of sample collection for the spring sampling campaign |
| sampling_doy_summer | Day of the year of sample collection for the summer sampling campaign |
| sampling_daytime_spring | Time of the day of sample collection (local time) for the spring sampling campaign. When a start and end time were given, the middle point is recorded. |
| sampling_daytime_summer | Time of the day of sample collection (local time) for the summer sampling campaign. When a start and end time were given, the middle point is recorded. |
| height_spruce1 | (Estimated) Height of spruce tree 1 [m] |
| height_spruce2 | (Estimated) Height of spruce tree 2 [m] |



| | |
|---|---|
| height_spruce3 | (Estimated) Height of spruce tree 3 [m] |
| height_beech1 | (Estimated) Height of beech tree 1 [m] |
| height_beech2 | (Estimated) Height of beech tree 2 [m] |
| height_beech3 | (Estimated) Height of beech tree 3 [m] |
| dbh_spruce1 | Diameter at breast height (DBH) of spruce tree 1 [cm] |
| dbh_spruce2 | Diameter at breast height (DBH) of spruce tree 2 [cm] |
| dbh_spruce3 | Diameter at breast height (DBH) of spruce tree 3 [cm] |
| dbh_beech1 | Diameter at breast height (DBH) of beech tree 1 [cm] |
| dbh_beech2 | Diameter at breast height (DBH) of beech tree 2 [cm] |
| dbh_beech3 | Diameter at breast height (DBH) of beech tree 3 [cm] |
| koppen | Three letter Köppen-Geiger climate code extracted from Beck et al. (2023). |
| canopy_cover_picture | Descriptor highlighting whether pictures of the canopy cover (see Table 4) are available in the WATSON_canopy_photos.zip file ("yes") or not ("no"). |
| canopy_cover | Mean canopy cover (C) for the sampling site, reflecting the average value for all photos for a sampling site (varying *n* per sampling site). Calculation of C as described in Section S3. |
| gap_fraction | Average gap fraction. One minus the average canopy cover, 1-C |
| network | Comment field, indicating to which monitoring network the site belongs |
| website_link | URL of a website describing the sampling site |
| paper_1 | DOI of paper 1 describing the sampling site |
| paper_2 | DOI of paper 2 describing the sampling site |
| paper_3 | DOI of paper 3 describing the sampling site |


**Table 3:** Description of the columns in the "WATSON_Isotopedata.csv" file containing all the isotope data and
additional information about the extraction [and units].

| Column name | Description |
|---|---|
| site_id | A three-letter identifier of the sampling site. Note that for the three sites (LIZ, GLS, WEI), an additional number was added indicating the species: "1" refers to beech and "2" to spruce. |
| country_id | A two-letter country code, as defined in ISO 3166-1 |

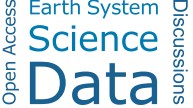

| | |
|---|---|
| sampling_date | Date that the sample was collected in yymmdd format |
| sampling_campaign | Descriptor indicating whether the sample was collected during the "spring" or "summer" sampling campaign. |
| sample_type | Descriptor indicating whether the sample was a "beech", "spruce" or "soil" sample |
| replicate | Number to indicate the tree from which the sample was taken (varying between 1 to 3, and occasionally between 4 to 6) or the replicate of the soil sample (typically only 1, but occasionall varying between 1 and 4). |
| spruce | Descriptor indicating if the sample was a vegetation sample from a spruce tree or if the soil was taken from a site that has spruce trees ("yes"), otherwise left blank |
| beech | Descriptor indicating if the sample was a vegetation sample from a beech tree or if the soil was taken from a site that has beech trees ("yes"), otherwise left blank |
| both | Descriptor indicating if the soil sample was taken from a site that has both beech and spruce trees ("yes"), otherwise left blank |
| species | Descriptor of the vegetation: "beech" and "spruce" for beech and spruce sites, respectively, or "both" if the soil samples were taken at a site where there are beech and spruce trees |
| soil_depth | Depth of the soil sample. Numbers ranging between 10 and 90, indicating the maximum depth of an interval, e.g. 10 for 0-10 cm, 20 for 10-20 cm, and 75 for 65-75 cm. For the vegetation samples, the field is left blank. |
| sample_id | A sample identifier used for all laboratory analyses |
| bark | "yes" when the sample included (remaining) pieces of bark, otherwise "no" |
| original_vial | The vial type in which the sample was received: exetainer that fit the cryogenic extraction line ("exetainer") or other types of gas-tight glass and plastic vials ("others") |
| extractionist | ID for the person responsible for cryogenic water extraction (A to D). Note that person D was only responsible for a very small subset. |
| cvd_slot_id | Slot ID of the cryogenic water extraction line at which a sample was placed during the extraction |
| exe_type | Numbers (1 to 10) indicate the type of exetainers (i.e., various combinations of glass vials, caps with rubber seals, and labels) |
| exe_weight | The mean weight of an empty exetainer of the exe_type, including glass vials, caps with rubber seals, and labels [mg] |
| fw | The fresh (field) weight of the sample [mg] |




| dw1 | The dry weight of the sample after cryogenic extraction [mg] |
|---|---|
| dw2 | The dry weight of the sample after cryogenic extraction and oven drying at 105°C for 24 h [mg] |
| awa | Absolute water amount extracted from the sample during cryogenic extraction [mL], calculated as: awa=(fw-dw1)/1000 |
| gwc | The gravimetric water content of the sample [%], calculated as: gwc = (fw-dw1)/dw1)*100) |
| tef | Total extraction efficiency [%], calculated as: tef = ((fw-dw1)/(fw-dw2))*100) |
| d18O | The $\delta^{18}O$ value (relative to VSMOW-2) as determined by the laser spectrometer [‰] |
| d2H | The $\delta^{2}H$ value (relative to VSMOW-2) as determined by the laser spectrometer [‰] |
| d18O_irms | The $\delta^{18}O$ value (relative to VSMOW-2) as determined by the isotope ratio mass spectrometer [‰] |
| d2H_irms | The $\delta^{2}H$ value (relative to VSMOW-2) as determined by the isotope ratio mass spectrometer [‰] |


**Table 4:** Description of the columns in the "WATSON_Canopydata.csv" file describing the canopy cover for the
sampling sites for which canopy pictures were available.

| Column name | Description |
|---|---|
| site_id | A three-letter identifier of the sampling site. Note that for the three sites (LIZ, GLS, WEI), an additional number was added indicating the species: "1" refers to beech and "2" to spruce. |
| country_id | A two-letter country code, as defined in ISO 3166-1 |
| species | Descriptor iinndicating the species for which the pictures were taken, either "beech" or "spruce" or "canopy" if the picture represents a picture of a mixed site or the overall canopy of the sampling site. |
| photo | Name of the file of the photo as given in the WATSON_canopy_photos.zip file. The general structure of each file name is: country_site_date_speciesm_xxx.JPG, where "country" indicates the country_id, "site" indicates the site_id, "date" the date that the picture was taken in yymmdd format, "species" the tree species (beech or spruce), "m" the tree number, and "xxx" refers to additional information, such as the distance from the tree in meters (1, 3, 5) or the direction in which the picture was taken (N, E, |



| | |
|---|---|
| | S, W). Where "canopy" is used for the "species", the picture shows the overall canopy of the forest site. |
| gap_fraction | One minus the canopy cover, 1-C |
| canopy_cover | The canopy cover (C), calculated as described in Section S3 [-] |


## 3    Results and discussion

### 3.1    Isotopic variation for the spring and summer sampling campaigns

The isotopic composition of the soil and the stem xylem water samples varied spatially (Figure 4). The samples
were more depleted in heavy isotopes at sites located further north and inland. Multiple linear regression analyses
showed that latitude, longitude, and elevation were all important variables explaining the observed spatial variation
in the isotopic composition of soil and stem xylem water (Table 5). Among the three geographic variables,
longitude and latitude explained most of the variance for seven of the eight cases shown in Table 5. Since the total
variance explained by latitude, longitude, and elevation was relatively low in most cases ($R^2 = 0.17$ to 0.6), other
factors likely contributed to the variation in the isotopic composition of the samples. In combination with the
gravimetric water content of the soil (e.g., "gwc"; Table 3), gridded climate data, and precipitation isotope data
(Nelson et al., 2021), the data could be useful for new soil and stem xylem water isoscape models or function as
additional data in hydrological studies.

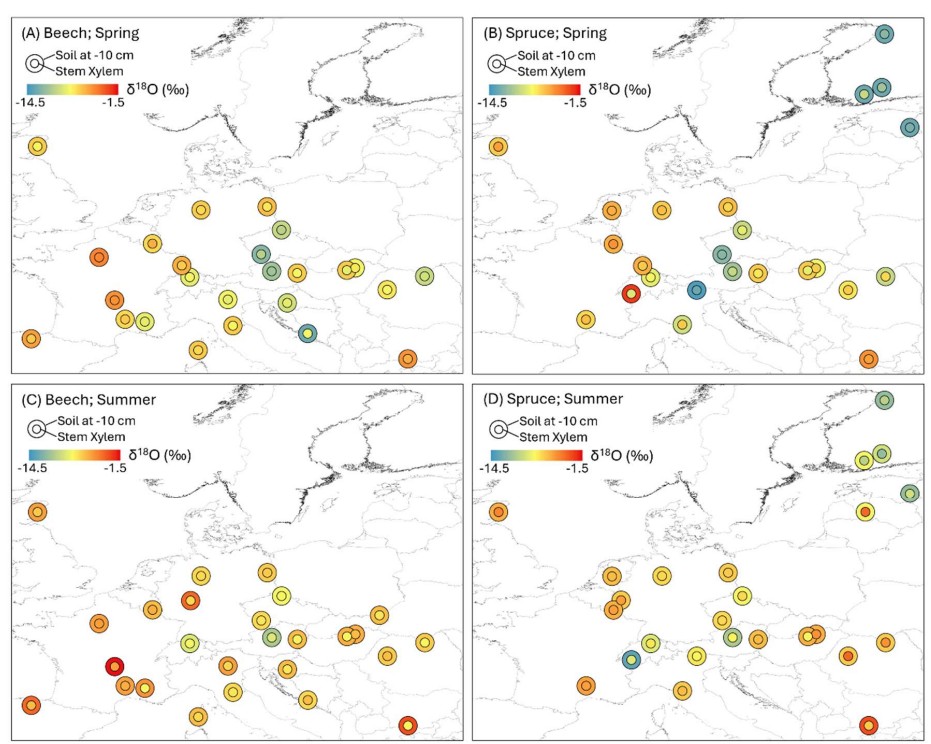


**Figure 4:** Map showing the δ¹⁸O values for stem xylem water (inner circle) and soil water at 0-10 cm (outer circle)
for the spring (A,B) and summer (C,D) sampling campaigns. Results for beech trees are reported on the left and
spruce trees on the right. For some sites, the isotopic composition of the stem xylem samples was similar to that
of the shallow soil (0-10 cm depth) (both circles have the same color); for others, the differences were large (i.e.,
the color of the inner and outer circle differs) indicating water uptake from a different (e.g. deeper) water source.

**Table 5:** Percentage of variance in δ¹⁸O values explained by latitude, longitude, and elevation, as determined by
multiple linear regression analyses. Values in bold indicate the highest relative contribution of a geographical
parameter to the total variance for each sample type for each campaign (Spring/Summer). $R^2$ reflects the total
variance explained by latitude, longitude, and elevation. All linear models were statistically significant (P < 0.001).

| Campaign | Sample | $R^2$ | Longitude (%) | Latitude (%) | Elevation (%) |
|---|---|---|---|---|---|
| Spring | Stem xylem (spruce) | 0.48 | 25 | 50 | 25 |
| | Stem xylem (beech) | 0.34 | 29 | 33 | 38 |
| | Soil (0-10 cm) | 0.35 | 50 | 38 | 12 |
| | Soil (30-90 cm) | 0.60 | 35 | 46 | 19 |
| Summer | Stem xylem (spruce) | 0.32 | 13 | 66 | 21 |
| | Stem xylem (beech) | 0.17 | 56 | 13 | 31 |



| | | 0.29 | 19 | 64 | 17 |
|---|---|---|---|---|---|
| | Soil (0-10 cm) | 0.29 | 19 | 64 | 17 |
| | Soil (30-90 cm) | 0.38 | 72 | 23 | 5 |


The isotopic composition of the soil and stem xylem water samples also varied between the two sampling
campaigns (Figures 4 and 5). For instance, $\delta^{18}O$ values were higher (i.e., less negative) in summer compared to
those of the spring for the different soil depths and the two tree species (unpaired t-test, P < 0.05), except for soils
in the depth range of 30-90 cm  for which there was no significant difference between spring and summer (unpaired
t-test, P > 0.05; Figure 5). For the $\delta^{18}O$ values of stem xylem water, the median seasonal difference (summer-
spring), averaged per site, was 0.8‰ across all spruce sites (ranging from -1.4 to 4.8‰) and 0.6 ‰ across all beech
sites (ranging from -1.9 to 2.9‰). In comparison, the average median seasonal $\delta^{18}O$ difference was larger and/or
showed higher a variability for soil water, e.g., 1.3‰ at 0-10 cm depth (ranging from -10.8 to 6.1‰) and 0.6 ‰ at
30-90 cm depth (ranging from -3.3 to 9.6‰). In spring, the $\delta^{18}O$ values of deep soils (30-90 cm) were only lower
(i.e., more negative) compared to those of the shallower soils (0-10 cm), while in summer, $\delta^{18}O$ values of deep
soils were lower compared to all other soil depths above 30 cm (unpaired t-test, P < 0.05). Similar seasonal
differences for stem xylem and soil water were observed for the $\delta^2H$ values (Figure 5). The data may, therefore,
be used to investigate the infiltration of precipitation and snowmelt into the soil, but also evaporative enrichment
of the shallow soil water, or to test models that simulate these processes.

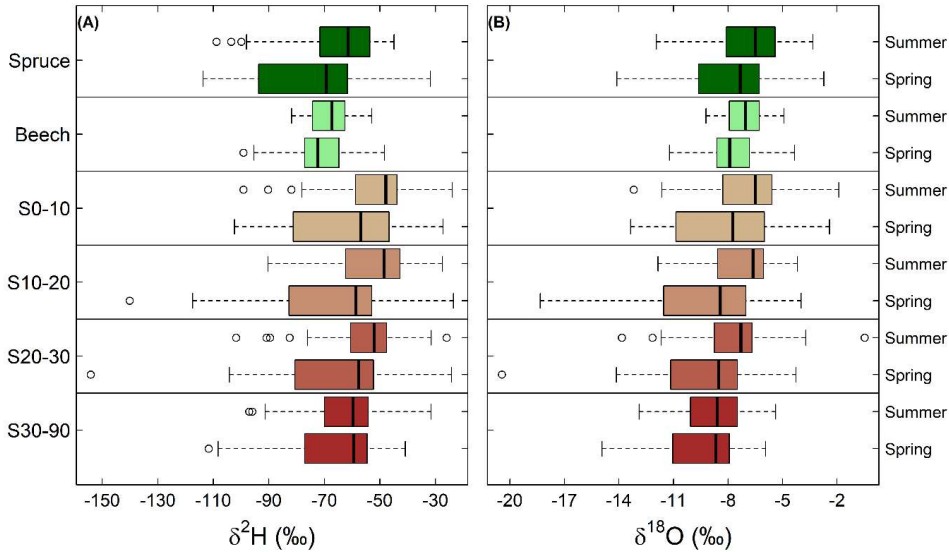


**Figure 5:** Boxplots for (A) hydrogen and (B) the oxygen isotopic composition ($\delta^2H$, $\delta^{18}O$) of stem xylem water of
both tree species (beech and spruce) and soil water at different depths for the spring and summer campaigns. Soil
depths are shown for 0-10 cm (S0-10), 10-20 cm (S10-20), 20-30 cm(S20-30) and 30-90 cm (S30-90). The vertical
line within the box indicates the median (50th percentile). The box represents the interquartile range (IQR),



spanning from the 25th percentile to the 75th percentile. The whiskers extend to the furthest data points within 1.5
times the IQR from the quartiles. Symbols outside the whiskers represent outliers.
Further, we found that the isotopic composition of the stem xylem water plotted in the range of soil water at the
site level ("overlap"), though not consistently across all sites (Figure 6). The mean $\delta^{18}O$ values overlapped for
more beech sites (68% in spring, 84% in summer) than for spruce sites (41 in spring, 48% in summer). The number
of sites for which the $\delta^{18}O$ values of the soil and stem xylem water overlapped was also larger for the summer than
for the spring sampling campaign. In contrast, the overlap in mean $\delta^{2}H$ values was higher for spruce sites (58% in
spring, 68% in summer) than beech sites (28% in spring, 23% summer). A lack of overlap may indicate that the
trees used water from other sources, such as recent precipitation events, water stored in organic surface layers,
deeper, unsampled soil layers or groundwater. Another explanation might be related to cryogenic water extraction
artefacts (see section on "Cryogenic water extraction biases").
The soil and stem xylem data could be used to test models that simulate plant-soil-water dynamics  (Klein et al.,
2014; Brinkmann et al., 2018; Knighton et al., 2020) and to test how this depends on site-, soil-, and tree-specific
information (Table 3). When the data are combined with isotope data of precipitation, such as those from the GNIP
network (e.g.,Terzer-Wassmuth et al., 2023), or models, such as Piso.AI  (Nelson et al., 2021), the data can also
be used to study the seasonal origins of tree water uptake, as well as the spatial and temporal patterns associated
with it (Allen et al., 2019; Floriancic et al., 2024a). For sites without overlap, the application of mixing models,
such as IsoSource (Phillips and Gregg, 2003) or MixSIAR (Stock et al., 2018), might be limited. However,
alternative mixing models with incomplete end-members could be tested (Kirchner, 2023).
For sites with both species, the isotopic data for the stem xylem water of the two species appear to be different
(Figure 6). The median difference between species across all sites for the mean $\delta^{2}H$ and $\delta^{18}O$ values (spruce-
beech), averaged per site, was 4.1‰ and 0.7‰ in spring and 10.1‰ and 1.1‰ in summer, respectively. Thus, the
stem xylem water in spruce tended to be isotopically enriched compared to ones in beech, which is consistent with
the generally shallower root system of spruce compared to beech. The data can therefore be used to study species-
specific differences in root water uptake depth across Europe.





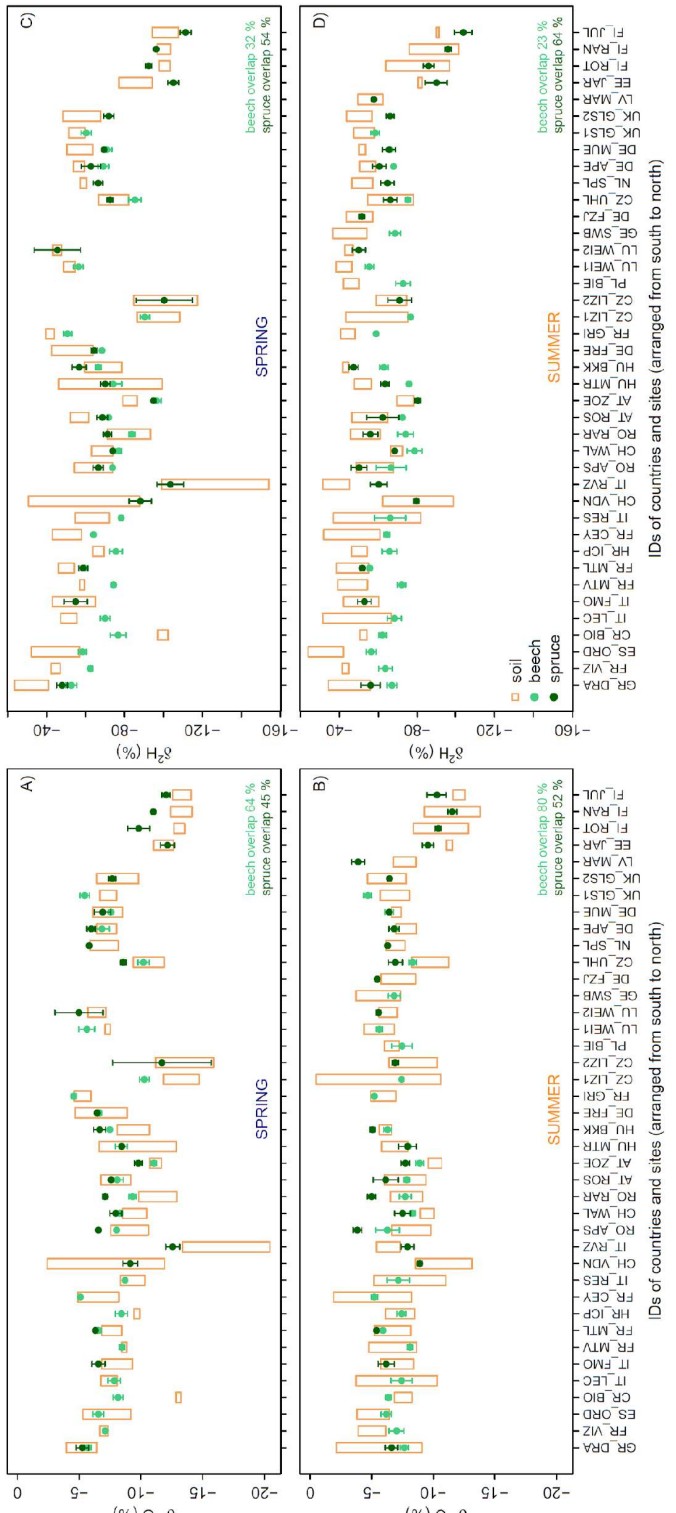

**Figure 6:** Overlap between the isotopic composition of soil and stem xylem water for spring (A, C) and summer (B, D) campaign. Oxygen ($\delta^{18}O$) and hydrogen ($\delta^{2}H$) isotope data are shown in the left and right panels, respectively. Orange bars indicate the minimum to maximum range of soil water isotope values. Mean values and standard errors are shown for the isotopic composition of stem xylem water.



### 3.2 Cryogenic water extraction biases

The dual isotope plots show that the isotope ratios of the soil were closer to the GMWL than those of stem xylem water for both species (Figure 7). However, particularly in summer, the isotope ratios of the shallower soils at some locations also deviated from the GMWL. This may indicate that the water in the shallow soil was affected by evaporation and that the trees used this enriched water. While evaporation might be responsible for some of the offset between the soil and stem xylem samples, there was no evaporative enrichment for most soil samples. Nevertheless, it should be considered that soil organic matter can bias the isotopic composition of the extracted water (Ceperley et al., 2024; Orlowski et al., 2016), as well as the presence of volatile organic compounds that may interfere isotopic analysis with laser spectrometers (Martín-Gómez et al., 2015). The latter, however, should be reduced by the use of the micro-combution modul in our study. Furthermore, given the relatively small isotopic differences between the laser and IRMS measurements (Figure 3), the overall large $\delta^2H$ deviation from the GMWL for the stem xylem samples is more likely caused by methodological issues related to the cryogenic vacuum distillation method (Chen et al., 2020; Diao et al., 2022; Barbeta et al., 2022). According to these studies, biases might be related to stem water content, differences in the isotopic composition of the xylem water and water in plant cells, exchange of H-atoms between organic material and water or water vapour, and isotope fractionation related to evaporation and sublimation during the extraction procedure.

To address these issues, we performed further quality checks for the cryogenic extraction (Figure 8). Although there was a significant difference in the total extraction efficiency for the samples handled by the three main lab technicians (one-way ANOVA, $P < 0.001$; Figure 8A), the efficiency did not depend on the cryogenic vacuum distillation slot (Figure 8B) and showed no systematic effect on the $\delta^2H$ and $\delta^{18}O$ values (Figure 8C). The presence of bark residue in the samples did not significantly affect the isotope signals (unpaired t-test, $P > 0.05$), although the slopes of the dual isotope plots tended to be different ($P = 0.06$, Figure 8D). Comparing the $\delta^2H$ and $\delta^{18}O$ values between samples stored in exetainers and other vials (Table 3, "original_vial") revealed no visual or statistical differences either, suggesting that sampling, transport, and transfer of samples from other vials to exetainers before cryogenic water extraction in the laboratory did not notably affect the isotope results. The data of this study can be used to further explore the cryogenic water extraction biases with the additionally provided site-, soil- and tree-specific information (Zhao et al., 2024; Sobota et al., 2024). Alternatively, they can be used to support other studies on methodological issues related to cryogenic water extraction.



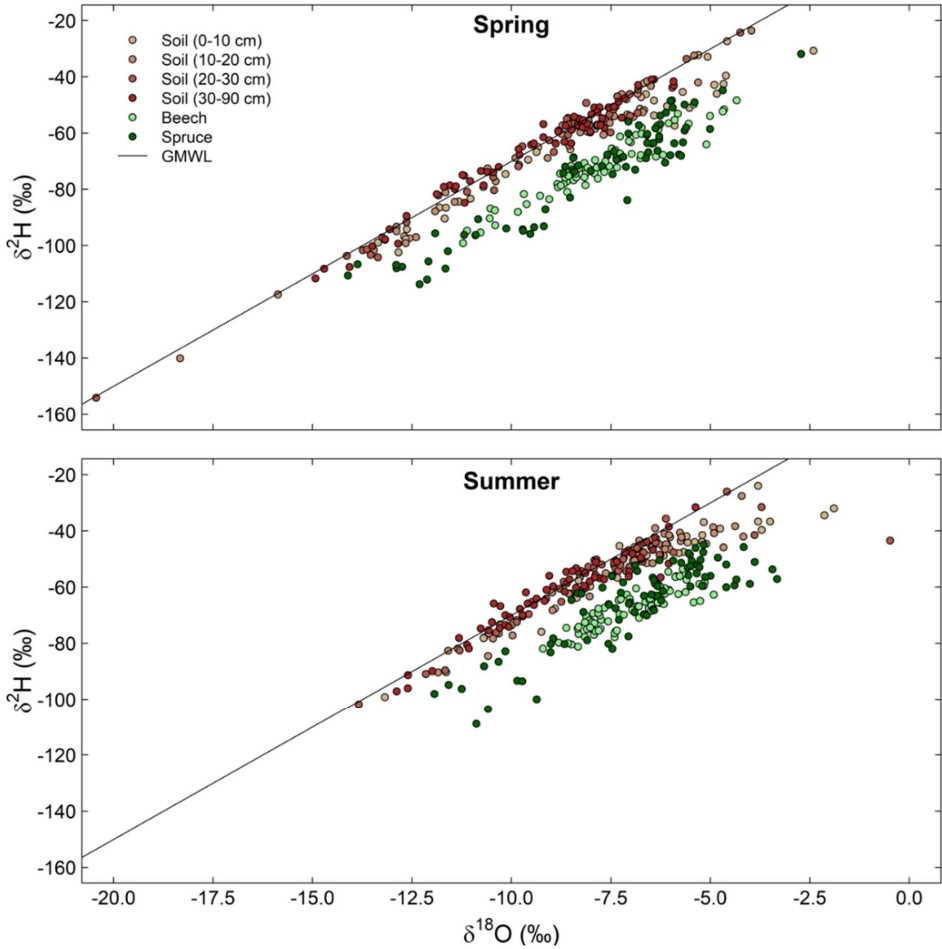


**Figure 7:** Dual isotope plots of oxygen and hydrogen isotope ratios ($\delta^2H$, $\delta^{18}O$) for all soil and stem xylem water
samples for the spring (top panel) and the summer (bottom panel) campaigns. Isotope values for soil samples are
color coded according to soil depth. GMWL = Global Meteoric Water Line: $\delta^2H = 8\ \delta^{18}O + 10$.



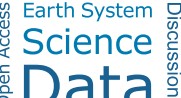

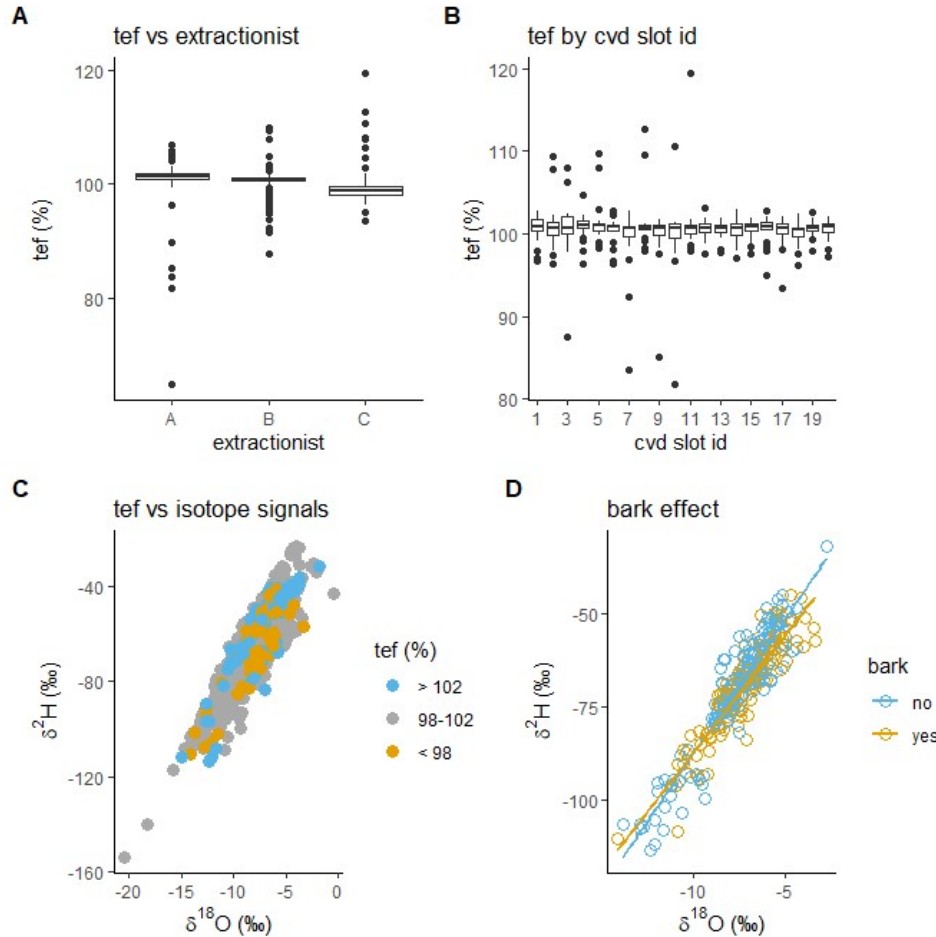


**Figure 8**: Total extraction efficiency (tef, %) quality checks: (A) tef values categorized by extractionist (Person A, B, or C) and (B) by cryogenic vacuum distillation slot IDs. Correlation between oxygen ($\delta^{18}O$) and hydrogen ($\delta^2H$) isotope values for (C) all samples colored by different tef categories and for (D) stem xylem samples with ("yes") and without presence of bark ("no"), including fitted trend lines.

## 4    Concluding remarks

We present a large pan-European dataset of soil and stem xylem water isotopes of two common tree species collected during spring and summer 2023. Since our observations are standardized according recently published sampling and extraction procedures (Ceperley et al., 2024; Scandellari et al., 2024), this data can serve as a baseline for future ecohydrological studies. This dataset is freely available and represents a valuable resource for different research topics. These may include identifying the factors that affect tree water uptake depth and the seasonal sources of water used by trees, calibrating and constraining isotope-aided ecohydrological models, incorporating



the data into isoscape models, or studying how biases caused by cryogenic water extraction vary by species, soil
type, or climate.

**Statistics**

For all statistical analyses we used R version 4.3.1 (R Core Team, 2023). For our multiple linear regression
analyses, we applied a cube root transformation to the data to address non-normality. We then utilized the R
package "relaimpo" (Grömping, 2006) to assess the relative importance of the geographical parameters in our
model. If data is presented for soil at a depth of 30-90 cm, it represents all available data points for soil depths
greater than 30 cm, without any additional modifications of the data.

**Funding**

This study was financially supported by the COST Action: "Water isotopes in the critical zone: from groundwater
recharge to plant transpiration - WATSON" CA19120 (www.cost.eu). The extraction of the water and isotope
analyses were financially supported by the Swiss National Science Foundation ("TreeWater", No. 205492;
"InsightForest", No. 213367) and by WSL ("Innovative project Oxygen17"). Alicia Correa was supported by the
German Academic Exchange Service (DAAD) from funds of Federal Ministry for Economic Cooperation (BMZ),
SDGnexus Network (No. 57526248). Aurel Persoiu and Andrei Popa were supported by UEFISCDI Romania (No.
PN-III-P2-2.1-PED-2019-4102 & No. PN-III-P4-ID-PCE-2020-2723). Maren Dubbert acknowledges the funding
by the Deutsche Forschungsgemeinschaft (No. 501530203) and by the Leibniz collaborative excellent grant (No.
K444/2022), supporting Alberto Iraheta, Charlotte Koeber, and Clara Rohde. Katja Rinne was funded by the
Academy of Finland (No. 343059). Research part at University of Oulu was supported by Research Council of
Finland (No. 347348 and No. 356043), and Marie Skłodowska-Curie Postdoctoral Fellowship (No. 101111527).

**Acknowledgements**

The pan-European sampling campaign and the data collection initiative was developed during a workshop of the
COST Action: "WATSON" CA19120 (http://www.cost.eu/; https://watson-cost.eu/) held in March 2023 in
Dubrovnik, Croatia. We thank Timon Dufner, Sophia Ezhold, Noemi Kammerlander, Alligin Gazhoul, Jan Ziegler,
Jonathan Frei, Roger Köchli, David Schweizer, Manuela Oettli for the laboratory assistance, as well as Enara
Aldai, Wisam Almohamed, Hatice Türk, Patricia Vieira Pompeu, Fernanda Gianasi, Konstantinos Voudouris,
Ionel Popa, Martine Helfer, Anna Meier, Ladina Gaudy, Laura Kinzinger, Dominik Gerber, Simon Bürki, Dominik
Dubach, Paavo Ojanen, Ellinoora Ekman, Christiaan van der Tol, and Joni Koivula for their help with site
selection, and/or sample- or metadata collection.

**Data availability**

All data is freely available under the agreement "Creative Commons Zero - No Rights Reserved (CC0 1.0)" in the
data repository EnviDat: Lehmann, M. M., Geris, J., van Meerveld, I., Penna, D., Rothfuss, Y., Verdone, M., Ala-
Aho, P., Arvai, M., Babre, A., Balandier, P., Bernhard, F., Butorac, L., Carrière, S. D., Ceperley, N. C., Chen, Z.,
Correa, A., Diao, H., Dubbert, D., Dubbert, M., Ercoli, F., Floriancic, M. G., Gimeno, T. E., Gounelle, D.,



Hagedorn, F., Hissler, C., Huneau, F., Alberto, I., Jakovljević, T., Kazakis, N., Kern, Z., Knaebel, K., Kobler, J.,
Kocum, J., Koeber, C., Koren, G., Kübert, A., Kupka, D., le Gall, S., Lehtonen, A., Leydier, T., Malagoli, P.,
Manca di Villahermosa, F. S., Marchina, C., Martínez-Carreras, N., Martin-StPaul, N., Marttila, H., Meyer
Oliveira, A., Monvoisin, G., Orlowski, N., Palmik-Das, K., Persoiu, A., Popa, A., Prikaziuk, E., Quantin, C.,
Rinne-Garmston, K. T., Rohde, C., Sanda, M., Saurer, M., Schulz, D., Stockinger, M. P., Stumpp, C., Vénisse, J.-
S., Vlcek, L., Voudouris, S., Weeser, B., Wilkinson, M., Zuecco, G., and Meusburger, K.: Soil and stem xylem
water isotope data from two pan-European sampling campaigns [dataset],
https://www.doi.org/10.16904/envidat.542, 2024.

## Competing interests

The authors declare that they have no conflict of interest.

## Author contribution (CRediT)

The WATSON sampling campaign core organization and writing team consisted of Marco M. Lehmann (MML),
Josie Geris (JG), Ilja van Meerveld (IvM), Daniele Penna (DP), Youri Rothfuss (YR) and Katrin Meusburger
(KM). Conceptualization: MML, JG, IvM, DP, YR, KM; Data curation: MML, MV; Formal Analysis: MML, JG,
IvM, DP, YR, MV, KM; Funding acquisition: MML, JG, IvM, DP, YR, KM; Investigation: MML, JG, IvM, DP,
YR, KM; Methodology: MML, JG, IvM, DP, YR, KM; Project administration: MML, JG, IvM, DP, YR, KM;
Resources: MML, KM; Validation: MML, JG, IvM, DP, YR, KM; Visualization: MML, JG, IvM, DP, YR, KM;
Writing – original draft: MML, JG, IvM, DP, YR, KM; Writing – review & editing: all.

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
