# Peer review of "Soil and stem xylem water isotope data from two pan-European sampling campaigns"

_Earth System Science Data, 2024_

## Author Comment (AC1)

We thank the reviewer for the useful comments and respond to them in blue font after the original comment in black font.

**Reviewer 1**

The manuscript "Soil and stem xylem water isotope data from two pan European sampling campaigns" presents a genuinely interesting data set as a result of an exemplary team effort. I can imagine that the presented data can help to answer some existing as well as to pose new questions regarding the investigation of tree water uptake with the help of stable water isotopes. I would recommend this manuscript to be accepted after minor revisions.

We thank the reviewer for the positive assessment of our manuscript. We agree that our isotope data are useful for an improved understanding of tree water uptake. Please find our point-to-point responses below.

The following issues should be resolved:

line 204: You requested 5 cm long sapwood samples. In the summer protocol you added 'Avoid sampling the heartwood'. Could you elaborate on why you added this to the protocoll and how well this instruction was subsequently followed by all contributors, especially the ones who sampled from trees with really small diameters (Tab 1 mentions minimum BHDs of 8 - 11 cm)? Do you think, that the isotopic samples of samples that included heartwood might skew your result? Do you think you could flag samples that might have inlcuded heartwood?

Thank you for this valuable comment. Although this aspect was mentioned in personal communications during online meetings before the first spring campaign, the core group decided to explicitly include the information (i.e. not to sample the hardwood) in the updated protocol for the summer sampling. In practice, one can often hear and feel when the heartwood is reached during coring, and we therefore think that in most cases the field crews correctly avoided it. We double-checked this aspect but could not reliably determine the presence of any heartwood after the water extraction, as dried samples lacked visible differences. As sapwood thickness varies with stem diameter and mature beech trees generally have thicker sapwood than mature spruce trees, small-diameter trees in our study, especially from spruce, may thus possibly include some heartwood despite efforts to avoid it. However, we decided not to flag individual samples as this could cause misinterpretation. Instead, we explicitly acknowledge this potential limitation for future data users and propose a solution in Section 2.3:

*The wood cores mainly reflect sapwood as participants were instructed to avoid sampling the heartwood because there are indications of isotopic differences between sapwood and heartwood (Fabiani et al., 2022). However, we cannot fully rule out the presence of heartwood in some samples as visual determination of the heartwood after water extraction was not possible. A heartwood correction based on mean wood core length and tree diameter could be developed. Such an adjustment may be particularly important for samples from smaller spruce trees, which are likely to have limited sapwood depth (Peters et al., 2019).*

line 250: An average gravimetric water content (gwc) of 40.9% seems pretty high for soil, especially if drier summer soil samples are included. For soils I am more familiar with typical values for volumetric water content (vwc), so maybe I'm just lacking an intuitive understanding of expectable gwc values. Could you explain why your average(!) gravimetric water contents ended up so high? Most of your soils are sandy loams, so if I'm not completely off with my gwc to vwc conversion, the average sample (including spring and summer) should have been very close to saturation. This does not seem right...

You are right that gravimetric water content (GWC) values should be interpreted carefully, in absence of direct volumetric water content (VWC) measurements. To clarify, GWC was calculated as GWC [%] = ((fw – dw) / dw) × 100, with fresh (fw) and dry weight (dw) measured in milligrams (but see Table 3). This is a standard mass-based approach to express the percentage of water relative to the dry mass of the soil. In the absence of any other site-specific soil moisture data like VWC, the GWC data can be used as a qualitative indicator of soil wetness.

To convert the GWC to a VWC, we need to multiply by the bulk density. We do not have this data but using an average number for woodlands across Europe of 0.73 g/cm$^3$ for 0-10 cm depth (Panagos et al., 2024, https://doi.org/10.1016/j.agee.2024.108907), we would get a VWC of 30%, which is perhaps closer to what the reviewer expected for the moisture content of the soil.

In addition, the high average value for soil GWC is partially influenced by the samples with high organic content for some sites but see also our response below. We now state in section 4:

*"In combination with the gravimetric water content of the soil as a qualitative indicator of soil wetness (i.e., "gwc"; Table 3), gridded climate data, and precipitation isotope data (e.g., Nelson et al., 2021), the data could be useful for new soil and stem xylem water isoscape models and be used as complimentary data in hydrological studies"*

Fig. 2C: Where do the really high gravimetric water contents > 200% come from? Are that measurement errors or is there another explanation for them?

These high gravimetric water contents are typical for organic soils, where the dry mass is very small compared to the amount of water retained. We now added in section 2.4:

*"The very high soil gwc values (> 200%) were all obtained for samples from the ROT site and reflect the high organic matter content (i.e., peat soil) for this site."*

Minor issues and some suggestions:

Fig.2: The caption refers to an inset in subfigure C, but the inset is shown in subfigure B

Thanks, we corrected the caption of Figure 2.

Fig.3: The legend item "Linear Model" in subfigure C confused me, since subfigure C does not contain such a line, but then I realized that this refers to the lines shown in subfigures A and B. Maybe you come up with a solution to improve this potentially confusing legend issue.

We removed "Linear Model" from the legend, as the black lines in Panels A and B clearly represent the linear regression lines.

Table 2: The aspect of the slopes could also be of interest. In case you have this kind of information, I suggest you add it to the data set.

Good idea, but unfortunately, we do not have the information on the slope aspect.

metadata.csv: Character encoding of the csv-files seems to be Latin3, my first guess of UTF-8 failed to properly display many of the special characters. Maybe add information on the proper encoding somewhere in the paper or within the repository.

Thank you for pointing this out. We have now updated all data files by converting them to UTF-8 encoding and standardized the delimiter to commas. This ensures that special characters are displayed correctly across platforms. We added a note in section 3:

*"All .csv files are encoded in UTF-8 and use commas as delimiters."*

Tab.5: The caption states that "Values in bold indicate the highest relative contribution...", but I do not see any bold values in the table...

Thanks for spotting this error. We now provide the corresponding values in bold.

Fig.6: The resolution of this Figure should be increased. The current version shows clear signs of compression artefacts. Better use a png-file, or even better better a vector graphic file format.

The available plots (including Figure 6) have been provided in screenshot quality to avoid large word and pdf files (as we had to share the document with many co-authors). Of course, we have high resolution plots for all figures, and we will provide them to ESSD in the final submission process to achieve the highest possible print quality.

line 74: The numbers in $\delta^2$H and $\delta^{18}$O should be in superscript.2

Thanks, corrected.

line 245-246: Apart from mean weight and maximum deviation, could you also specify the standard deviation?

Okay, we have modified the sentence and replaced the maximum deviation (= 0.3) by the standard deviation, now stating:

*"The average weight of the exetainers was 13.0 ± 0.2 g (SD)."*

line 271: "with [the] laser spectrometer"

Thanks, corrected.

line 277: "offset[s] between"

Thanks, corrected.

line 352: "than for spruce sites (41[%] in spring, 48% in summer)"

Thanks, corrected.

line 418: "standardized according [to] recently published"

Thanks, corrected.

---

## Author Comment (AC2)

We thank the reviewer for the useful comments and respond to them in blue font after the original comment in black font.

**Reviewer 2**

The authors present an interesting dataset of soil moisture and plant xylem water (beech and spruce) isotopic compositions measured across 40 sites in the spring and summer seasons. The dataset documentation is clear and transparent. The potential uses of this dataset are also immediately clear to me. I fully support publication of this data release. I provide some minor comments below:

We thank the reviewer for the positive assessment of our manuscript and the dataset. Please find our point-to-point response below.

Specific comments:

Line 74: The "2" and "18" should be superscript.

Thanks, corrected.

Line 99: perhaps consider "resistance and resilience". Deeper roots can help a plant recover after accumulating damage during drought, but deeper roots can also help individual trees avoid negative consequences during a drought in the first place.

We agree. We have now added "*resistance*" to the text.

Line 99: rather than pointing to the topic of "climate change" it could be more informative to list the changing hazards that impact trees where greater RWU would help to avoid or mitigate negative consequences (i.e., drought, fire). Climate change just seems to broad a term in this application. Ocean acidification falls under climate change but probably doesn't impact trees much.

Okay. To narrow the scope of the word "climate change", we now link climate change specifically to hazards such as the increasing frequency and intensity of droughts in the following sentence, which we see as the most important factor affecting RWU. Now stating:

*"Understanding how tree water uptake from soils varies with species, site characteristics, time, and across climate zones is essential to assess forest resistance and resilience to climate change; particularly the response of forests to the increasing frequency and intensity of droughts"*

Line 102: rather than "determination" I would suggest "estimation" as these isotope methods are constantly debated and refined. This paper and dataset highlight potential issues that still should be resolved.

Good point, we changed the text accordingly.

Line 110: and possibly (iv) that all end-members have been measured with proper consideration for the locations of uptake and the transit time of water through the plant. The article states in the discussion the possibility of water sources beyond those that were measured.

We agree. We have now modified point (i) to consider your comment and changed it to:

*"Additionally, it is assumed that: (i) the sampling design captures all end-members with a proper representation of the spatiotopic variability of their isotopic composition"*

Line 111-112: Water uptake is just one dimension of plant responses to drought though, right? Stomatal closure, xylem resistance to embolism, canopy position, internal water storage, etc. etc. etc. Roots are certainly important, but they do not paint a complete picture of drought resistance and resilience. As an example, cacti generally have shallow roots relative to forest trees but are capable of existing in much more arid regions.

Yes, root water uptake is just one component of a multifaceted drought response. Physiology and hydraulic traits are also key for plant drought responses. We have slightly expanded the sentence and now state:

*"the method can either independently or in combination with other measurements (e.g., in combination with assessment of physiological or hydraulic traits) be used to …"*

Line 130: Possibly this is the only international network that explicitly states this as the primary goal. The LTER and CZO networks across the US measured soil and plant xylem water isotopic compositions at almost all of the sites similar to this study. The political boundary of the US makes it so that all exist in one country so they are not "international", but the US is comparatively large by land area, spanning a diverse range of ecoregions. My point is that there have been other attempts at something similar, while this sentence makes it sound like this has never been done before. While the CZO is inactive (and LTER is threatened), NEON sites remain active, and isotopes are being measured at some.

Thanks, that is highly interesting, as we were not aware of these attempts. We deleted "only" and added nuance to the statement:

*"The Moisture Isotopes in Biosphere and Atmosphere (MIBA) network, initiated by the IAEA in 2003-2004, is a rare example of an international network to survey the isotopic composition of water across different ecosystem compartments…"*

As a sidenote, we are fully aware the US is almost as big as Europe, but we also think sampling across multiple countries is more challenging than sampling across the US, e.g., due to issues with the shipment and import of samples, languages, etc., see also response to comment above.

Lines 174 – 175: It would be really interesting to share the lessons learned from these calls. My research group is constantly communicating with others regarding challenges in field sampling, storage, shipping, analysis, etc. This could be very helpful. Possibly this is already all documented in the Ceperley et al (2024) paper?

That's a good point, which was also mentioned by reviewer 3. While large parts on sampling and extraction procedures are covered by Ceperly *et al.* (2024), we expanded the concluding remarks:

*"Establishing this data set with a geographic cover across Europe was feasible because the participants took advantage of an EU Cost Action with members in most European countries. We believe that limiting the number of samples to 6 to 8 per site contributed considerably to the success of the data collection. Centralizing the laboratory and analytical work avoided potential inter-laboratory biases, while the availability of an import license reduced shipping times and lowered the risk of sample loss."*

Line 189: Maximum of all sites, or the maximum at each site?

We refer to the maximum of all sites, which is now stated:

*"The maximum soil depth varied between 0.3 m and > 1 m. For half of the sites was the maximum soil depth > 0.6 m."*

Line 189: Maybe clarify "canopy cover" here. I work with foresters who use lidar to construct 3D models of canopy structure. They might not call photos representative of canopy cover.

Yes, there are more sophisticated methods to measure canopy cover, but we feel that the sentence already describes the method explicitly so that users can decide whether they want to use it or not. Still, we replaced "determined" by "estimated" to provide more nuance to our statement:

*"Canopy cover was estimated for 30 of the 40 sampling sites from non-hemispherical photographs taken systematically at varying distances from the stem with a smartphone camera (Supplementary materials S3)."*

Fig 1: The country borders are showing up at a low resolution. Is there a way to improve this? Minor issue.

The available plots (including Figure 1) have been provided in screenshot quality to avoid large word and pdf files (as we had to share the document with many co-authors). Of course, we have high resolution plots for all figures. In the higher resolution plots, the country borders are more clearly visible. We will provide them to ESSD in the final submission process to achieve the highest possible print quality.

Line 202 – 203: I'm a little confused by the terminology. I typically think of "stem" as a < 3 cm diameter branch. Here it sounds like you are coring the trunk if you used an increment borer. Is it stems or trunks?

From our experience, "stem" is often used when it comes to "trunk" or "tree-ring" samples and we therefore think that this is a meaningful term. This is also underlined by having no comments on this from any of the 70 authors of the manuscript before the submission process and a quick informal check with other colleagues. Nevertheless, we now added "tree" to the title to make it more obvious that the dataset deals with samples from trees. We also added on several occasions that the stem xylem samples were taken from the trunks of trees in the abstract, introduction and material & methods. We also added a definition in section 2.3:

*"Thus, in this study, "stem" refers specifically to the trunk of the tree, excluding branches and other aboveground components."*

Line 205: was heartwood removed too?

We instructed the field crew to avoid sampling heartwood. Often one can hear and feel when the heartwood is reached so we think that in most cases the field crews did this correctly. That said, we could not determine the absence or presence of heartwood from the dry wood samples after water extraction and therefore cannot fully exclude that some samples may have contained some heartwood, particularly spruce samples from small trees. Please see the detailed response to a similar comment from reviewer 1. We have added some sentences to make this more explicit in section 2.3:

*"The wood cores mainly reflect sapwood as participants were instructed to avoid sampling the heartwood because there are indications of isotopic differences between sapwood and heartwood (Fabiani et al., 2022). However, we cannot fully rule out the presence of heartwood in some samples as visual determination of the heartwood after water extraction was not possible.*

*A heartwood correction based on mean wood core length and tree diameter could be developed. Such an adjustment may be particularly important for samples from smaller spruce trees, which are likely to have limited sapwood depth (Peters et al., 2019)."*

Line 211: Why say "typically"? There was a protocol or not? What are those "other depths"?

"Typically" because most samples were taken at the indicated depth, i.e., according to the protocol. "Other depths" refer to soil samples that were taken at depths that deviate from the protocol (e.g., 65-75 cm as stated in "soil_depth" in Table 3) because the maximum soil depth varied among sites. We rephrased the sentence to be clearer:

*"The samples were taken from a single soil core at three to five depths, typically at 10 cm intervals (0-10, 10-20, 20-30, 50-60, and 80-90 cm below the surface). In some cases, other depths were sampled, or the sampling interval was 20 cm. The number of soil samples and the depth of the deepest soil sample depended on the soil properties (e.g., rocky soils) and the maximum soil depth at the sampling location."*

Line 214: How close?

Unfortunately, we do not know the exact distance between the location of the soil core and the trees for each site. It was only instructed that the location should be between the selected trees as mentioned in section 2.1.

The Goldsmith et al (2019) paper and my own sampling has me convinced that we need soil samples at least in triplicate due to high spatial heterogeneity. Was this disregarded to simplify data collection?

We are aware of the paper (now cited) and the issue. We discussed it during the initial phase of the project. However, collecting more soil samples would have substantially increased the workload for water extraction and isotope analysis. To ensure feasible sampling, shipping, and participation across many sites, we reduced the number of soil samples to a practical minimum. This decision is now reflected in the concluding remarks:

*"We believe that limiting the number of samples to 6 to 8 per site contributed considerably to the success of the data collection."*

Interestingly, at the few sites where multiple soil cores were collected, we observed a high similarity in the isotopic composition across soil depths, supporting the adequacy of our sampling strategy.

Line 217: was root mass with depth measured at any sites? Wouldn't the root profile be informative for designing the soil sampling protocol of each site?

We agree that this would be very helpful information. The data exist for some sites, which can be deduced from "website_link" or "paper_x" in Table 2. However, because root traits are not easy to assess and need some expertise to establish them correctly (and are difficult to determine based on just one soil core), we did not include root traits in the sampling design and protocols, also to mitigate the risk of "losing" potential contributors.

Line 222: How fast is "back in the laboratory"? We've done tests where we put samples into plastic containers and leave them at room temperature and stored cold for varied amounts of time. We observe no problematic effects for the first 6 hours, but we do often observe problems for samples held at longer storage times. This depends entirely on the container and ambient

environmental conditions, but it might be good to quantify the potential issues related to sample storage and transport time.

While we know the date and time of the sampling in the field (mean ~12:30 local time), we have not noted the arrival of the samples in the laboratory. We asked and therefore must assume that all contributors stored their samples in the refrigerator in the afternoon/evening of the sampling day. Because most of the analyzed samples were kept in gas-tight "Exetainers" from Labco during storage and shipment, which we have used for several decades for various studies, we doubt that evaporative effects caused isotope fractionation during sample transport. This is supported by our finding that samples handled/shipped in different plastic or glass vials showed no signs of evaporative isotopic enrichment due to water loss (new Figure 8E). We now state this more clearly in section 4.2.:

*"In contrast, vial type significantly interacted with sampling campaign (P < 0.001), with no effect in spring but a more depleted signal for the vial type "others" compared to "exetainer" for the summer sampling campaign. This pattern provides no indication of evaporative isotopic enrichment resulting from sample handling during the warmer summer conditions."*

Still, we agree that it will be interesting to see what happens to the samples during longer periods of storage, but this is beyond the scope of this study.

Line 223: "shipped without cooling" and "4 weeks" is concerning to me. Did you perform a test to confirm that the shipping time didn't result in problems? You could take a dried plant or soil sample, spike it with a standard, hold it for the duration, and then extract to estimate the potential bias.

We decided for "Shipment without cooling" to avoid complicated and costly shipment of the packages, again to mitigate the risk of "losing" potential contributors. "4 weeks" is the maximum time it took. Most samples were handled and arrived much faster. We also minimized the shipping duration with an import license, as now stated in the concluding remarks:

*"Centralizing the laboratory and analytical work avoided potential inter-laboratory biases, while the availability of an import license reduced shipping times and lowered the risk of sample loss."*

We do not think that we can design an easy experiment that would be representative of all the variability that might be related to the various steps between sampling, shipment, CVD extraction, isotope analysis, and different soil types across ~70 authors and 40 sites across Europe. In other words, even if we would generate one estimate for an isotopic effect due to for example shipment period, we cannot be sure if it is representative for individual samples as we did not track the exact timing of the individual steps between sampling and isotope analysis. We therefore think that testing this uncertainty will not result in a meaningful outcome and is beyond the scope for this study.

However, the vial type used for sampling ("original_vial", Table 3) can, in a broader sense, serve as a test case for potential uncertainties related to storage and shipping. While most samples were collected in Exetainers, approximately 15% were shipped in other containers ("others"), including plastic or glass vials of differing shapes and materials. These alternative vials were not specifically certified as gas-tight and therefore posed a higher risk of evaporative isotope fractionation during storage and transport. The absence of a detectable vial-type effect for spring and depletion of heavy isotopes for samples store in the "others" compared to "exetainer" for the summer sampling campaign suggests negligible evaporative influences (i.e. water loss) during sampling, shipping, and the transfer of material into Exetainers prior to

cryogenic water extraction. This further supports the robustness of our dataset in this context. We added this information to section 4.2:

*"In contrast, vial type significantly interacted with sampling campaign (P < 0.001), with no effect in spring but a more depleted signal for the vial type "others" compared to "exetainer" for the summer sampling campaign. This pattern provides no indication of evaporative isotopic enrichment resulting from sample handling during the warmer summer conditions. Given that the "others" vial type comprises only ~15% of samples, spread across no more than 8 of 40 sites in both campaigns, we consider this effect unlikely to confound the overall dataset, though it may warrant consideration in future analyses. Collectively, these results support the overall reliability of the dataset and its suitability for analyses of cryogenic water extraction biases and methodological evaluation"*

Line 248: Does the final dataset exclude samples outside of the acceptable ranges, or does it include all values?

No data was removed. Users can use their own threshold and filter data for "tef" values outside of the optimal range, if they want to exclude those samples.

Line 371: Even if they were similar, they could still be used to study differences. It might just produce a null result, or maybe environmental conditions varied such that its interesting that the values were similar. More data is always good!

Agreed, we rephrased the last sentence of the paragraph to provide a broader and more generic perspective:

*"The observed isotopic variability in stem xylem water among species and sites suggests that both species-specific differences in root water uptake depth and the environmental drivers of root water uptake across Europe can be inferred from these data."*

Line 400: What if the effect was similar across all containers and transportation routes? This doesn't rule out the problem for me. A small experiment to estimate the bias attributable to containers/transport might be worthwhile to at least rule this out entirely.

While we agree that this is interesting, it is beyond the scope of the paper describing the dataset and its potential uses. See our response to similar comments above.

---

## Author Comment (AC3)

We thank the reviewer for the useful comments and respond to them in blue font after the original comment in black font.

**Reviewer 3**

"Soil and stem xylem water isotope data from two pan European sampling campaigns" presents a very interesting collaborative dataset and provides a well-written and generally clear documentation of the data and sampling protocols. The usefulness of the data and potential for applications are clear and exciting. Upon addressing minor comments, I recommend the publication of this manuscript and data.

We thank the reviewer for the positive assessment of our manuscript. We are similarly excited to use the data for further publications and hope that others will use it as well. Please find our point-to-point response below.

Specific Comments:

Line 191: When photos were taken for both campaigns, were they averaged? Or was one selected over the other? I now see this is answered in Tables 2, but it would benefit from mentioning here too.

Okay, we have now added more details for Canopy cover in the Table 1 caption: *"\*\*\* based on the average value for all photos for each sampling site"*

Section 2.3: Where were soil samples collected at each of the sites? Was there a protocol for location relative to the trees sampled? I think this is described more in the protocols document, but it should be expanded on here, too.

In the sampling protocols we stated: *"Take a maximum of 5 soil samples at 0-10, 10-20, 20-30, 50-60, 80-90 cm at a location between the 3 (or 6) trees that you sampled"*, which is similar to the description in the manuscript. We do not have more information to describe the location per site any better. We slightly modified the sentence and now state:

*"In addition to the stem xylem samples, soil samples were taken with a manual soil auger at a location between the selected trees."*

Line 211: It seems that samples were taken between 30-50cm based on the protocols. Why is this not a range mentioned here?

For soil samples below 30 cm, the variability of the sampled soil depth varied due to differences in the maximum soil depth (and the presence of rocks). To avoid adding too many details for individual sites, we decided on a more generic statement and refer to "other depths". We modified the sentence:

*"The samples were taken from a single soil core at three to five depths, typically at 10 cm intervals (0-10, 10-20, 20-30, 50-60, and 80-90 cm below the surface). In some cases, other depths were sampled, or the sampling interval was 20 cm."*

Line 215: Can you add text to clarify if these soil samples were taken at deeper depths or at separate locations?

We added more details to the sentence:

*"At some sites and during certain campaigns, soil samples were also taken from two to three additional nearby locations (up to four in total), resulting in a varying number of samples and sampling depths."*

If separate locations, were these averaged for subsequent analyses or are they all included as separate data points?

The data of the different locations (= auger holes) are provided as separate data points.

Line 223: Can you expand on why they were shipped without cooling? It's stated in the immediately prior sentence that refrigeration helps reduce moisture loss and evaporation/fractionation.

Indeed, cooling can mitigate evaporative water loss, which causes unwanted isotopic fractionation during storage. It will also reduce microbial growth and slow the decomposition of organic material. However, we decided against shipment of the samples in refrigerated conditions to avoid complex and costly transport logistics. That said, we minimized shipment duration using an import license. Now stated in concluding remarks:

*"while the availability of an import license reduced shipping times and lowered the risk of sample loss"*

Line 249: I find this sentence challenging to follow. I assume the 40.9% values is referring to all samples, but it's not immediately clear and makes the "respectively" description at the end feel disconnected from the beech and spruce.

We rephrased the sentence to be clearer:

*"The gravimetric water content ("gwc") varied among sample types and averaged 41% for soil, 61% for beech xylem, and 84% for spruce xylem samples (Figure 2C)."*

Table 2: exe_type Are these exe_types listed and described somewhere? I think it would be beneficial to understand the different types of containers used. If this is listed somewhere, please add a reference to the location.

Good point. We now provide various weights and details on the variable "exe_type" in the supplement section S4. This list reflects quite some work, and therefore we think that it is useful that these details are published somewhere.

Line 310: I think it would be beneficial to add a sentence somewhere in the text (probably Introduction) on how these variables are related to isotopic processes to motivate these models.

That is a good idea. We now state in the introduction:

*"This includes geographic details, information on soil type, texture and maximum depth, details on forest stands, tree diameter and height, sampling information, as well as data on canopy cover/gap fractions as indicators for stand density and tree health and crown defoliation (Bussotti et al., 2024). Together, the metadata and isotope data provide a strong foundation for future research on tree water use, model testing, and isotope mapping."*

Table 5/Section 3.1: Is there a reason the soil samples were grouped in this way, and the 10-20 cm and 20-30cm samples were excluded? In the text, soils are suddenly grouped and discussed

as "shallow" (0-10 cm) and "deep" (30-90cm). Please add a sentence (perhaps in Section 2.3 or Paragraph 2 of 3.1) where these groupings of the soil are clearly defined and briefly discussed.

Good point. There was no specific reason. To be consistent we now add the details for the soil depth 10-20 cm and 20-30 cm to Table 5 too. We also revised the paragraph and now avoid the groupings "shallow" and "deep" and refer to the actual soil depth. But see our response to the comments below as well.

Line 331: So δ18O values were only higher in summer compared to spring for the shallow soil depth (0-10cm), not for all soil depths? The first part of the sentence seems to say that δ18O is higher in summer for both shallow and deep ("the different depths"), but then it seems to indicate that this is only true for shallow soils.

We revised the sentence to be clearer:

"$\delta^{18}O$ values were higher in summer compared to those of the spring for stem xylem water of both species and for soil water at 0-10 cm, 10-20 cm and 20-30 cm (unpaired t-test, $P < 0.05$). The $\delta^{18}O$ values of soil water at depths of 30 to 90 cm did not differ seasonally (unpaired t-test, $P > 0.05$; Figure 5)."

Line 337-339: What is the difference between "shallow" soils and soils above 30 cm? Is it that summer deep samples are more negative than the 0-10 cm & 10-20cm & the 20-30cm soils? While the spring samples are lower than the 0-10cm samples but not the 10-20cm or 20-30cm samples? It's hard to tell if these analyses are only comparing the shallow and deep groups of samples or the 10 cm increments as described in the methods.

Indeed, that was confusing. We now avoid using "shallow" and "deeper" categories and revised the sentences to be clearer:

 "Comparisons across all soil depths shows that in spring, site-level mean $\delta^{18}O$ values of soil water at 30–90 cm depth were lower (i.e., more negative) compared to those at 0–10 cm (unpaired t-test, $P < 0.05$) but not to those at 10-20 cm or 20-30 cm (unpaired t-test, both $P > 0.05$). In contrast, in summer $\delta^{18}O$ values at 30–90 cm depth were lower than those at 0-10, 10-20 and 20-30 cm (unpaired t-test, $P < 0.05$). Similar seasonal differences for stem xylem and soil water were observed for the δ2H values (Figure 5)."

Line 368: Similar to line 249. There are several sentences throughout that are tough to follow because there are two seasons, two analytes, and/or two or more subsets of samples mentioned within the same sentence, and it's not immediately clear which grouping of the samples "respectively" is referring to.

We agree and now see that the sentence structure caused confusion. We rephrased the sentences to:

"Our data also shows a clear isotopic difference in stem xylem water between the two tree species (Figure 6). The mean species difference (spruce-beech) in $\delta^2H$ and $\delta^{18}O$ values across all sites was 5.5‰ and 0.8‰ in spring and 9.5‰ and 1.1‰ in summer, respectively."

In addition, we further revised the overall manuscript and rephrased overly long and complicated sentences in the manuscript.

Line 371: Please add a citation

We now cite Goldsmith *et al. (*2019) as a good example of differences in root water uptake for spruce and beech.

4 Concluding Remarks: I understand this may be out of the scope, however, I feel a short section highlighting some key "lessons learned" regarding sampling, analysis, and collaborative efforts across this impressive geographic extent would be beneficial and of interest to the community.

Good point, we added some lines to the concluding remarks: *"Establishing this data set with a geographic cover across Europe was feasible because the participants took advantage of an EU Cost Action with members in most European countries. We believe that limiting the number of samples to 6 to 8 per site contributed considerably to the success of the data collection. Centralizing the laboratory and analytical work avoided potential inter-laboratory biases, while the availability of an import license reduced shipping times and lowered the risk of sample loss."*